



# Snow farming: Conserving snow over the summer season

Grünewald Thomas[1], Lehning Michael[1,2], and Wolfsperger Fabian[1]

[1]WSL Institute for Snow and Avalanche Research SLF, Flüelastrasse 11, 7260 Davos, Switzerland
[2]Cryos, School of Architecture, Civil and Environmental Engineering, École Polytechnique Fédérale de Lausanne, GRAO 402 – Station 2, 1015 Lausanne, Switzerland

*Correspondence to:* Thomas Gr"unewald (gruenewald@slf.ch)

**Abstract.** Summer storage of snow for winter touristic purpose has seen an increasing interest in the last years. Covering large snow piles with materials such as sawdust enables to conserve more than two thirds of the initial snow volume. We present detailed mass balance measurements of two sawdust covered snow piles obtained by terrestrial laser scanning during summer 2015. Results indicate that 74% and 63% of the snow volume remained over the summer. If snow mass is considered instead of volume, the values increase to 85% and 72% which is attributed to settling and densification of the snow. Additionally, we adapted the one-dimensional, physically based snow cover model SNOWPACK to perform simulations of the sawdust covered snow piles. Model results and measurement agreed extremely well at the point scale. Moreover, we analyzed the contribution of the different terms of the energy balance to snow ablation for a pile covered with a 40 cm thick sawdust layer and a pile without insulation. Shortwave radiation was the dominant source of energy for both scenarios but the moist sawdust caused strong cooling by long-wave emission and negative sensible and latent heat fluxes. This cooling effect reduces the surface energy balance by a factor or 12. As a result only 9% of the net shortwave energy remained available for melt. Finally, sensitivity studies of the parameters thickness of the sawdust layer, air temperature, precipitation and wind speed were performed. We show that sawdust thickness has a tremendous effect on snow loss. Higher temperatures and wind speeds increase snow ablation but are less important. No significant effect of additional precipitation could be found as the sawdust remained wet during the entire summer. However, switching of precipitation of completely would strongly increase melt.

## 1 Introduction

Snow storage or snow farming is the conservation of snow during the warm season of the year. Purposes of summering snow or ice are diverse. Already in ancient time, ice and snow were stored for cooling of food and houses (Taylor, 1985; Skogsberg and Nordell, 2001; Morofsky, 2007). Another example for a traditional application of snow storage is the collection of snow in deep underground wells e.g. in Afghanistan (Bhattacharyya et al., 2004). The water stored in the snow could be used for irrigation or as drinking water during summer. Rising energy costs have increased interest in snow or ice as cooling source (Nordell, 2014). In several regions of the world, such as Scandinavia, Northern America, China or Japan snow storage has seen a revival for air conditioning of buildings (Skogsberg and Nordell, 2001; Nordell and Skogsberg, 2007; Morofsky, 2007; Hamada et al., 2010, 2012) or food cooling (Kobiyama et al., 1997; Nordell, 2014): During winter, large amounts of natural or machine-made snow are collected and stored. During summer the melting snow provides cooling energy for air conditioning.





Another application of snow conservation is the protection of glaciers from melt. At some ski resorts, thin textile covers, called geotextiles have been used to blanket critical areas of the glacier surface at the end of the winter season (Olefs and Lehning, 2010). The properties of the covering material, such as albedo, thermal conductivity, emissivity or permeability influence the energy balance of the snow resulting in reduced ablation (Olefs and Obleitner, 2007; Olefs and Lehning, 2010).

Olefs and Fischer (2008) performed field tests at two Austrian glaciers where they analyzed different types of geotextiles and plastic fabrics as covering material. They found that ablation could be reduced by up to 60% in comparison to a natural snow surface. Finally, Olefs and Lehning (2010) used the snow cover model SNOWPACK to simulate the effect of geotextiles on energy balance and snow ablation on glaciers. They showed that most of the performance of geotextiles is attributed to increased short wave reflectance compared to the dirty firn surface. Thermal insulation and cooling by latent heat transfer

contribute but are less important. Olefs and Lehning (2010) summarize that geotextiles are very effective covering materials for glaciers but were not suitable at lower elevations where turbulent fluxes typically dominate the energy balance.

The idea to conserve snow for touristic purposes, also at lower elevations, came up in Scandinavia more than a decade ago. Similar to the previously mentioned snow storage applications, large amounts of snow are collected at the end of the winter or produced by snow machines and conserved over the summer months. In the following autumn, the remaining snow is used

as basis for the preparation of winter sport facilities such as cross-country tracks, ski-runs or ski-jumps. A frequent motivation of winter sport destinations to store snow is the hosting of large sports events, such as cross-country races in autumn or early winter. The conserved snow provides a guaranty for a basic amount of snow for track preparation, independent of the current weather conditions. Examples for such events are the Cross-country Skiing World Cup in Davos, Switzerland, the Biathlon World Cup in Östersund, Sweden or the Ski-jumping World Cup in Titisee-Neustadt, Germany. The largest application of snow

storage was performed for the winter Olympics held in Sotschi, Russia in 2014. About $800.000\,m^3$ of snow were preserved as reserve for the preparation of alpine ski-racing tracks in case of lack of snow (Lintzen, 2016). Other motivations are the early preparation of training facilities for athletes or an early opening of ski slopes for the public. Some ski resorts collect snow at the end of the winter and conserve it at neuralgic spots, such as terrain breaks or depressions. In the succeeding winter the remaining snow is used for leveling of these depressions.

Different organic materials, such as sawdust, chipped wood, cutter shaving, bark mulch or straw have been used for the coverage of the snow piles (Skogsberg and Lundberg, 2005). Moreover, non-organic fabrics such as styrofoam plates, polymeric foam, geotextiles and combinations of different materials have been applied. All these materials act as insulating layers that reduce heat transfer from the atmosphere to the snow. Depth, thermal conductivity and heat capacity are the key characteristics for their insulation performance. In addition, most organic materials are able to store significant amounts of moisture.

Evaporation of the water cools the surface and reduces snow ablation (Skogsberg and Lundberg, 2005). Moreover, the high shortwave reflectivity (albedo) of some cover materials reduces solar energy input. Obviously, meteorological conditions, such as air temperature, solar radiation, moisture, precipitation and wind strongly affect snow melt. First investigations (Rinderer, 2009), basic calculations (Skogsberg and Nordell, 2001) and reports from practitioners indicate that the amount of snow that is lost during summer is in the range of 20 to 50%. However, mass balances based on accurate measurements have not yet been

published.



In recent years the number of snow-conservation projects and sites has strongly increased, especially in Scandinavia but also in the Alps. Snow farming can be seen as an adaption strategy to climate warming but is also required to satisfy customers demands of an early winter season start. However, direct economic effects, such as increasing numbers of tourists in the early winter season are probably limited (Dreier, 2010). The main benefit for the destinations will rather be related to higher media

presence and positive image in relation to general snow security of the destination and to hosting professional winter-sports events such as word-cup races independent of the weather.

Detailed mass balances on this application of snow storage are currently not available. The only study known to the authors is a field test by Rinderer (2009) who observed two snow heaps at a location near Davos, CH (1620 m a.s.l.). One was covered with geotextile (4 mm) and the other with a 40 cm layer of sawdust. Based on a simple analysis of time lapse photography Rinderer

(2009) concluded that the first pile nearly melted completely while the latter lost about 30% of its volume. Additionally, the different cover layers were implemented into the physically based snow cover model SNOWPACK (Lehning et al., 2002a). Simulations of the field experiments showed plausible results. But due to the limited quantification of the volume losses a precise validation of the model results were lacking.

This paper presents the first scientific publication on snow farming for touristic purposes as described above. Contrary to

Rinderer's relatively rough estimation, we present detailed measurements of the volume loss based on high-resolution terrestrial laser scanning surveys. In 2015 measurement campaigns have been performed at two sites, at the same site as Rinderer's study (Davos) and in the Martell valley of South Tyrol. These measurements for the first time allow for a detailed analysis of the volume loss of such snow farming applications. We present an analysis of the small-scale spatial characteristics of snow ablation at the two snow heaps. The study is completed by simulations using the SNOWPACK model. Model runs of for the

two sites are performed to evaluate the ability of SNOWPACK to reproduce the evolution of the snow piles and to analyze the impact of the different terms of the surface energy balance. Finally, a sensitivity study, assessing effects of different settings and changed atmospheric conditions is performed. The respective results are discussed and summarized in the end of the paper.

## 2   Methods and data

### 2.1   Study sites

Observations of snow-farming projects at two study sites in the Alps are presented. The first is in the Flüela valley near the city of Davos in the Eastern part of Switzerland (Fig. 1). The storage site is a forest clearing at the valley bottom located at 1620 m a.s.l. near to the mouth of the Flüela valley (lat: 46.808°, long: 9.868°). The direction of the valley is west to east and the site is shadowed by mountains, in particular by a 2500 m high mountain in the South.

Snow farming has been operated at the Flüela site since 2008. A large snow pile is formed by machine made snow produced

during the winter months. The snow pile also contains relatively small amounts of natural snow resulting from precipitation events during winter. In spring this pile is smoothed and covered by a 40 cm layer of sawdust. The purpose of snow farming in Davos is to provide a short track for cross-country skiing that is usually opened by the beginning of November. The track





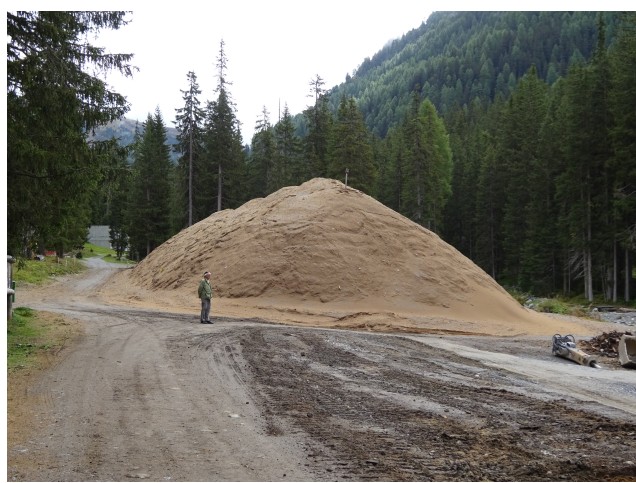

**Figure 1.** Flüela snow heap after coverage with sawdust in autumn 2015

serves as training facility for professional athletes and also provides the snow basis for the cross-country skiing world cup in early winter.

Meteorological data from two different automatic stations were analyzed for the Flüela site. The station "Windtunnel" measuring air temperature (TA), relative humidity (RH), wind speed (VW) and direction (DW) is installed at the roof of a

concrete building next to the snow heap. The height of the sensors is about seven meters above ground which is similar to the height of the snow pile. The remaining meteorological parameters required as input for the simulation (incoming shortwave radiation (ISWR), incoming longwave radiation (ILWR) and precipitation (P)) were taken from weather station DAV that is located at a similar elevation (1596 m a.s.l.) in the town of Davos about 2 km away and has high quality meteorological observations. The local effect of terrain shadowing to incoming shortwave radiation was corrected by applying a shade-filter

available in the pre-processing library for meteorological data, MeteoIO (Bavay and Egger, 2014). This filter transfers measured radiation of a measurement station to another site taking into account local shading.

No automatic station had been set up directly on top of the snow pile in 2015. However, a simple weather station had been placed from 27 May to 17 July 2016 at the top of the snow heap. These observations could be applied to assess the representativeness of the sensors and to correct the input data used for the simulation. Moreover, the long record of meteorological data

at DAV (since 1998) enables to characterize the summer 2015 in comparison to the climatic mean.

The second snow pile was in the Martell valley of South Tirol (Fig. 2). The storing site is a forest clearing at 1710 m a.s.l. at the bottom of the valley (lat: 46.517°, long: 10.720°). Up to 3700 m high mountains surround the valley that opens to the Northeast. The snow dumping site is a concave, northerly exposed slope with steepness rising from 0 and 50°. Work flow and purpose of the pile are similar to Flüela: Snow is produced with snow machines during the cold season and covered in

spring. Different to Flüela, the covering material at Martell also contained a portion of larger wood chips. In 2015 3.5 km of cross-country tracks could be prepared with the stored snow. Same as in Davos, the aims of snow farming are the provision





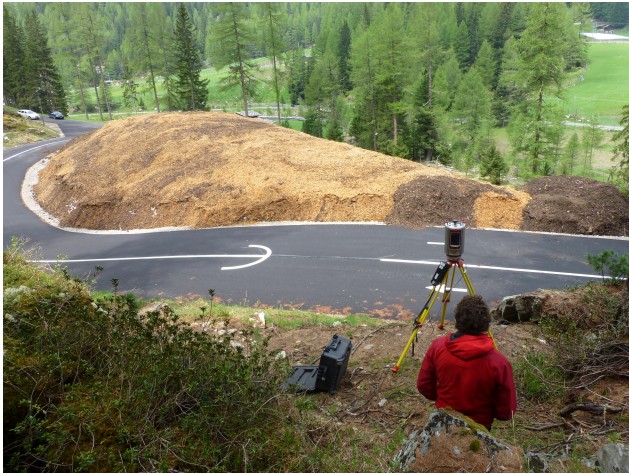

**Figure 2.** Martell snow heap covered with sawdust and wood chips in spring 2015. The Riegl VZ6000 laser scanner can be seen in the foreground.

of early training possibilities for athletes and a guarantee of basic snow amounts, independent of weather conditions during autumn and early winter.

We used meteorological observations of the automatic weather station Hintermartell (Autonome Provinz Bozen, 2016b) to assess meteorological forcing. This station is only 250 m away and is equipped with sensors for TA, RH, P, ISWR, VW and

DW. Measured ILWR is not available in the region and was therefore parametrized (see Sect. 2.4).

As the station was only established in 2009, we employed data from the slightly higher (1851 m a.s.l.) and 1.2 km distant station "Stausee Zufritt" (SZ) (Autonome Provinz Bozen, 2016a) to evaluate the climate conditions of summer 2015 in a climatological context.

### 2.2   Field measurements

### 2.2.1   Terrestrial laser scanning

Measurements of the snow volume stored at the end of the winter season and of the volume that remained in autumn were performed by terrestrial laser scanning (TLS). TLS proved as a highly accurate method to obtain digital surface models. Changes of surface volumes can easily be obtained by subtracting of two succeeding surveys. Repeated TLS was successfully applied to calculate snow volumes in high spatial resolution and with a vertical accuracy of less than 10 cm (e.g. Prokop et al.,

2008; Grünewald et al., 2010; Revuelto et al., 2014).

We used a Riegl VZ6000 terrestrial laser scanner (Riegl Laser Measurement Systems GmbH, 2015) shown in Figure 2. The VZ6000 operates in the near-infrared spectral range and is characterized by high accuracy (15 mm), precision (10 mm) and a beam divergence of only 0.12 mrad. Measurement frequencies of 300 kHz were used. For this study, the maximum measurement distance of the target area was less than 150 m. The respective beam width for this distance would be 33 mm.





**Table 1.** Estimated volumes of snow remaining in the Martell dump.

| Zone | Snow depth [cm] | Area [$m^2$] | Volume [$m^3$] |
|------|-----------------|--------------|----------------|
| 1    | 10-20           | 1290         | 129-258        |
| 2    | 50-100          | 407          | 204-408        |
| 3    | 150-200         | 170          | 255-340        |
| Sum  |                 | 1867         | 588-1006       |

Because of this rather small measurement range the maximum angular step width (0.002°) could be reduced to 0.03°. To overcome scan shadows and to capture the complete snow piles TLS measurements from six (five) positions were taken at each survey day at the Flüela (Martell) snow pile. TLS surveys were performed at the Flüela site at 29 April and 8 October 2015. These two data sets allow for the calculation of the volume change of the snow heap during summer. In order to determine
the absolute volume and by this the relative loss of the snow pile a bare ground (without snow pile) elevation model would be required. Such a model was not available and could not be surveyed as the site was always covered, either by the snow heap (during summer) or by the stored sawdust (after snow removal). However, the site is a quite homogeneous, only slightly sloped flat that could well be approximated by a plane that was interpolated from the edges of the snow pile. Note that at the time of both surveys the snow pile was covered by a layer of sawdust. 23 manual measurements of the sawdust depth were
performed in two cross transects with an avalanche probe perpendicular to the slope. The mean depth was 32 cm (standard deviation 11 cm) corresponding to a depth of about 38 cm when calculated in vertical (gravitational) direction.

At the Martell field site TLS measurements were performed at 19 May and 28 October 2015. Similar to Flüela these dates indicate the beginning (few days after the pile was covered) and the end of the melt season (briefly before the snow was turned out to the cross country track). Identical to the Flüela data set the period between the two surveys corresponds to 163 days
but both surveys were obtained three weeks later. Different to Flüela, the covering material was not sawdust but wood chips. Particle size of these wood chips varied from 0.1 mm (like sawdust) to pieces with a size of several centimetres. It needs to be considered that additionally the surface of the pile was covered by a thin layer of snow during the second survey that resulted from an early snow fall. We calculated a mean snow height (HS) of 6 cm (standard deviation 1.5 cm) from 27 manual HS measurements. During data processing, this mean value was applied to remove the additional volume caused by the snow
cover. As the wood chips layer was hard frozen, its thickness could not be measured with a penetrating probe but, based on the roughly known total volume of wood chips (800-1000 $m^3$) deployed it was estimated to be about 35 to 45 cm (Martin Stricker personal communication). As mentioned before, the ground surface of the dump is not flat at Martell but characterized by variable sloped topography. Therefore, a third TLS survey was performed at 19 November. Even though most snow had been turned out at that time, some snow still remained in the dump. Unfortunately, HS could not be measured with a probe as the
snow was extremely hard frozen. We therefore had to estimate the remaining snow amount based on our visual impression: The surface was visually classified to three zones that were then mapped with a differential GPS. The respective values are listed in Table 1. Based on these estimations we used a mean value of 800 $m^3$ for the corrections.



### 2.2.2  Snow density

Snow densities are on the one hand required for the initialization of the SNOWPACK model and on the other hand for the transformation from snow volume to snow mass or snow water equivalent (SWE).

Snow densities were measured at the beginning (Apr 2016) and at the end of the storage periods (Oct 2015) at different
locations and depths of the Flüela snow heap. Five snow volumes were extracted with a core driller providing cylindric snow samples ($d = 72\ ,mm$) of different length. Ten volume samples were taken with a $100\,dm^3$ cubic density cutter. Additionally, three cubic snow samples were taken to determine densities by X-ray micro-computer tomography (Heggli et al., 2011).

### 2.3  Data processing

TLS requires substantial efforts of data post-processing which was done with the commercial software RiSCAN Pro provided
by the manufacturer (Riegl Laser Measurement Systems GmbH, 2011). In a first step the extremely large amount of data needs to be reduced. This was done by removing all data outside of the area of interest and by aggregating the remaining data points to a 5 cm 3D grid (octree filter). Next, the data points from the multiple scan positions need to be combined. For this registration a set of eight (seven) reflector tiepoints that were installed at fixed locations such as walls of buildings in the surrounding were used. Global coordinates of the reflectors were recorded with differential GPS and total station. In a second step an
optimization function called Multi Station Adjustment was applied to further improve the registration of the scan positions (Riegl Laser Measurement Systems GmbH, 2011). Finally, the data of each survey were transformed to the global coordinate system (CH1903 and UTM respectively) by applying the global coordinates of the reflectors and the merged and filtered point clouds were exported. ESRI ArcMap 10.2 was used to rasterize the data. First, data were triangulated and data gaps that were partly existing, especially at the crone of the pile were closed by linear interpolation. Finally, raster data with a resolution of
10 cm were composed and used for volume calculations and further analysis.

### 2.4  Snowpack model

SNOWPACK is a one-dimensional physically based snow model which has been developed to simulate state and evolution of the snow cover. The multi-layer model rests upon a detailed description of the energy and mass fluxes within the snow and between snow, atmosphere and soil (e.g. Lehning et al., 2002a, b; Wever et al., 2015). For this publication SNOWPACK
was applied to simulate the evolution of the two snow heaps during a complete summer season. Aims were to identify the dominant processes related to mass conservation during snow farming and to evaluate the models applicability to improve and plan existing and upcoming snow farming projects.

SNOWPACK is initialized with a snow profile, describing the state of the snow at the beginning of the simulation. With the one-dimensional SNOWPACK model, we chose to simulate ablation at the top of the respective piles. Corresponding to the
maximal HS measured at the two heaps, HS were set to 8.60 m for Flüela and 7.20 m for Martell. The snow was subdivided to ten homogeneous layers of fully rounded grains with a grain size of 1 mm and a density of $553\,kg/m^3$. These values are typical for settled machine made snow and are based on the measurements described earlier. Moreover, the snow was assumed



**Table 2.** Initial properties of the sawdust layer as applied in SNOWPACK. The values are based on investigations of Rinderer (2009).

| | |
|---|---|
| Volume fraction of soil [%] | 50 |
| Volume fraction of water [%] | 20 |
| Volume fraction of void [%] | 30 |
| Density of soil [$kg/m^3$] | 245 |
| Bulk density [$kg/m^3$] | 323 |
| Thermal conductivity [W/mK] | 0.07 |
| specific heat capacity [J/kgK] | 990 |
| Spectral albedo [%] | 0.5 |

isothermal (snow temperature 0°C) holding a portion 3% of liquid water. Temperature and water content were not measured directly but are based on our impressions during density sampling. Two layers of soil were added to the snow profile. One at the bottom (ground) and one at the surface representing the covering material. A 40 cm thick covering layer representing the amount of sawdust obtained from measurements at the two snow heaps is entitled as reference simulation in the following.

Characteristics of the sawdust layer are listed in Table 2. As we assume similar properties of sawdust and the mixture of sawdust and wood chips used in Martell, same model settings were used for all simulations.

SNOWPACK is driven by the meteorological measurements TA, RH, VW, DW, ISWR, ILWR and P. These parameters were obtained from nearby automatic weather stations as described earlier. As no measurements for ILWR were available for the Martell region, it was approximated using a combination of a clear-sky parametrization (Dilley and O'Brien, 1998)) and an

all-sky algorithm (Unsworth and Monteith, 1975) as described in Schmucki et al. (2014). This parametrization was also applied to account for changes in long wave radiation that go along with increasing temperatures (Schmucki et al., 2014) as simulated in the sensitivity runs with higher air temperatures (Sect. 3.5).

All input data were filtered, quality checked and resampled to 30 min time steps using the meteorological input-output library MeteoIO (Bavay and Egger, 2014). The modeling time step was set to 15 min. Model outputs are time series of snow profiles and fluxes reflecting the state of the snow pack at different points of time. In context of this study, snow mass, density, water

content and the respective energy and mass fluxes are the parameters of interest. In addition to the reference simulation with measured meteorological data and a 40 cm sawdust cover, sensitivity studies were performed by varying the meteorological input parameters and the depth of the covering surface layer (Table 3.5). For each run, only one of the parameters was changed while the measured input was used for the others (as described above). We performed seven sensitivity runs for different depths

of the covering material by increasing it from 0 to 60 cm. In addition to the reference simulation, we performed two model runs for wind speed, three runs for precipitation and four runs for temperature as shown in Table 3.5. Note that also ILWR was adapted for the temperature simulations as described above. Sensitivity runs are on the one hand valuable to identify the most relevant impact factors that can help to improve settings for snow farming projects and on the other hand to test the operability of snow farming as presented to other sites or to future climatic conditions.



**Table 3.** Parameters and settings (offset or factor) applied for the sensitivity study.

| Parameter | Offset | Factor |
|---|---|---|
| Depth of covering layer [cm] | -40,-30,-20,-10, +10,+20 | |
| Air temperature [°C] | -1, +1,+2,+5 | |
| Wind speed [m/s] | | 0*, 2, 5 |
| Precipitation [mm] | | 0, 2, 5 |

* Constant wind speed of 0.3 m/s as hard-coded in SNOWPACK

## 3 Results

### 3.1 Meteorological data

Half-hourly meteorological measurements of VW and TA obtained by the stations at the top of the Flüela snow pile and the station Windtunnel were analysed for the period 27 May to 17 July 2016. Air temperature at Windtunnel was highly correlated ($R^2 = 0.9$) and showed a low bias of 0.4°C with measurements at the pile. Contrary to TA, VW showed stronger deviations. A mean VW of 0.3 m/s measured at Windtunnel indicates a clear underestimation of the wind at the pile (mean = 0.9 m/s). This might be attributed to wind sheltering effects of local topography and surrounding trees. Based on the measured data we calculated a linear regression function ($VW = 2.1 * VW_{WT} + 0.3$, $R^2 = 0.6$) that was then applied to correct VW for the simulations.

Meteorological input data as applied for the simulations are presented in Figure 3 and 4. Note that the data were aggregated to daily values for the plot while half-hourly values were used as input for the simulations. Mean temperatures of the survey period (29 Apr to 8 Oct) were 11.3°C at Windtunnel and 10.6°C at Hintermartell (19 Mai to 27 Oct). The mean of corrected VW at Windtunnel (WT) was 0.86 m/s and the measured VW at Hintermartell (HM) was 1.3 m/s. Precipitation cumulated to 543 mm (HM: 534 mm). Net shortwave radiation summed to 410 $kJ/m^2$ (HM: 337 $kJ/m^2$) and longwave radiation was calculated -276 $kJ/m^2$ (HM:-166 $kJ/m^2$). Lower temperatures and irradiation is mainly attributed to the survey period at Martell that was three weeks later. This is confirmed by a comparison of data for an identical period (1 April to 30 September 2015) that demonstrated very similar climatic conditions at the two sites (WT: 10.1°C /, 514 mm; HM: 10.2°C, 514 mm).

Temperature and precipitation data of the nearby weather stations DAV and Stausee Zufritt (SZ) with long-term records were analyzed to rate the summer 2015 in relation to climatic mean values. At DAV the mean temperature of the summer half year (1 Apr to 30 Sep) 2015 was 10.1°C (SZ: 9.3°C) and therefore 0.8°C (SZ: 0.4°C) warmer than the mean of the same period of the last 15 years. Precipitation cumulated to 571 mm (SZ: 409 mm) in comparison to a 15-year mean of 645 mm (SZ: 434 mm). It can therefore be concluded that the summer half year 2015 was warmer and drier at both sites, while the difference was more pronounced at Flüela. This is in agreement with the findings of climate reports for Switzerland (MeteoSchweiz, 2016) and South Tirol (Munarni et al., 2015) that rated the year 2015 as the warmest and the summer as the second warmest since start of the measurements.

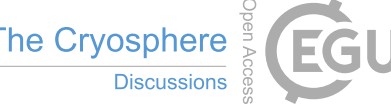

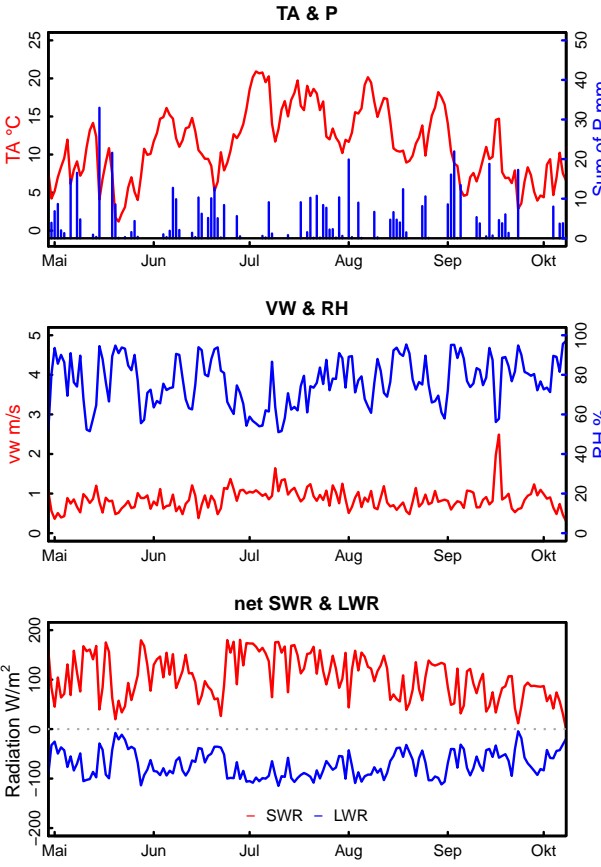

**Figure 3.** Meteorological input data aggregated to daily values for the Flüela site. Air temperature (TA) and precipitation are shown in the upper panel, wind speed and relative humidity (RH) in the middle and net shortwave (SWR) and longwave (LWR) radiation in the lowest panel. Measurements were obtained from the stations WT (TA, VW, RH) and DAV (SWR, LWR, P).

### 3.2 Snow density

Four of the five density samples collected at the snow pile in spring could be used for micro CT analysis. One sample had been destroyed during preparation. Mean density calculated from the samples with a volume of about $5\,cm^3$ was $555\,kg/m^3$ (standard deviation: $18\,kg/m^3$).

5    Measurements obtained in autumn showed a mean density of $556\,kg/m^3$ (n=6, standard deviation: $15\,kg/m^3$) at the position nearest to the snow surface (about 3.7 m above ground and 1 m below the snow surface) increasing to $578\,kg/m^3$ (n=6, standard deviation: $8\,kg/m^3$) about 2.1 m above ground and to $681\,kg/m^3$ at the lowest sample position (n=1). The resulting bulk density (mean of all three levels) was $606\,kg/m^3$. This is in accordance to a densification of about 0 to 23% (9% for the mean values) in relation to the initial values.



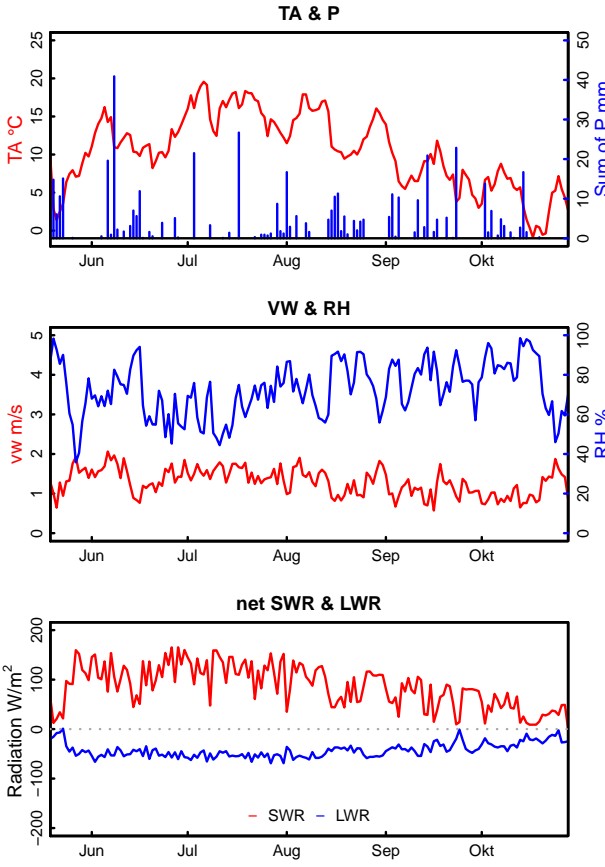

**Figure 4.** Meteorological input data aggregated to daily values for the Martell site. Air temperature (TA) and precipitation are shown in the upper panel, wind speed and relative humidity (RH) in the middle and net shortwave (SWR) and longwave (LWR) radiation in the lowest panel. Measurements were obtained from the stations HM and LWR was parameterized as described above.

### 3.3 Measured snow volume and mass balance

This section presents results obtained from the TLS surveys. The focus of the analysis is on the Flüela data set. Values for Martell are provided in brackets.

Snow-height maps calculated from the TLS data of the two snow heaps are presented in Fig. 5c,d , Fig. 6c,d and the respective values are shown in Table 4. The maximal height of the Flüela heap was 8,99 m (Martell: 7,61 m) in spring and decreased to 7,86 m (Martell: 6,37 m) in autumn. Mean HS (including sawdust layer) reduced from 4.07 m (Martell: 3.33 m) to 3.15 m (Martell: 2.18 m) and the mean change in snow height (dHS) was 2.70 m (Martell: 2.32 m).

Red colours in Fig. 5e and indicate that dHS was highest at the crown of the snow pile. The rather steep crest that was present in spring clearly leveled during summer resulting in the formation of a small, few meters wide plateau, starred with several





**Table 4.** Geometric properties and snow depth statistics of the Flüela and the Martell snow heap as calculated from the TLS surveys.

| | **Flüela** | | | **Martell** | | |
| --- | --- | --- | --- | --- | --- | --- |
| | 29 Apr | 8 Oct | Difference | 19 May | 28 Oct | difference |
| 3D surface area [$m^2$] | 2087 | 1992 | | 2150 | 2094 | |
| Ground area [$m^2$] | 1685 | | | 1913 | | |
| Volume [$m^2$] | 6862 | 5307 | 1555 | 7138* | 4820** | 2318 |
| Relative loss [%] | | | 22.6 | | | 32.5 |
| $HS_{mean}$ [m] | 4.07 | 3.15 | 0.92 | 3.33 | 2.18 | 1.15 |
| $HS_{max}$ [m] | 8.99 | 7.86 | | 7.61 | 6.37 | |
| $HS_{sd}$ [m] | 2.64 | 2.25 | 0.52 | 2.14 | 2.0 | 0.39 |
| $dHS_{max}$ [m] | | | 2.70 | | | 2.32 |
| $dHS$ at $HS_{max}$ [m] | | | 1.35 | | | 1.25 |
| Relativ loss at $HS_{max}$ [%] | | | 15.1 | | | 16.6 |

*800 $m^3$ (remaining snow) added to raw data. ** 800$m^3$ (remaining snow) added and 126$m^3$ (fresh snow) removed. Details are described in Sect. 2.2.1.

small depressions. In contrast, the dark blue sections visible at the edges of the pile in Fig. 5c and 6c represent areas where height increased. This is, however, an artefact that can be explained by relocation of sawdust, caused by gravitational and man-made relocation and by the fresh snow at Martell. Low dHS values at the edges can also partly be explained by smaller snow heights, naturally limiting dHS to this initial value. Moreover, red colours near to the edge of the Martell pile point at higher

snow ablation in these locations (Fig. 6e). These higher dHS might be attributed to increased melt caused by the proximity to the paved road (Fig. 2): It can be expected that the road heated much stronger due to low albedo and thermal properties and thus contributed additional energy by lateral advection of heat towards the snow heap (Mott et al., 2013, 2015). Besides this no eye catching patterns such as a possible influence of exposition are visible. This impression is confirmed by Fig. 7a that shows boxplots relating dHS to aspect. Fig. 7b, however, indicates a negative relationship of dHS and slope. Higher ablation seems to

happen in flatter places. Lower wind speeds over steep slopes might be a possible explanation.

    Hillshade images and maps of HS and dHS are displayed in Fig. 6 for the Martell snow pile. Note that HS and dHS shown in Fig. 6c,d,e were not corrected for remaining snow in the depot and for fresh snow covering the surface during the survey in autumn. The reason is that only few measurements or estimations of the correction values were available that might well be used for reasonable adjustment of mean values or total snow volumes but are inadequate for correction in high spatial

resolution. Complementary to lower HS at the edges mentioned earlier, this fact, explains many of the negative HS at the edges of the heap (Fig. 6c,d) and negative dHS in Fig. 6e. Contrary to Flüela, the geometry of Martell heap was much flatter and no such remarkable changes in surface characteristics, such as the formation of a plateau could be detected (Fig. 6a,b). Same as for Flüela aspect seems not to significantly influence dHS (Fig. 7c). Moreover, Fig. 7d, does not suggest a clear relationship between dHS and slope.



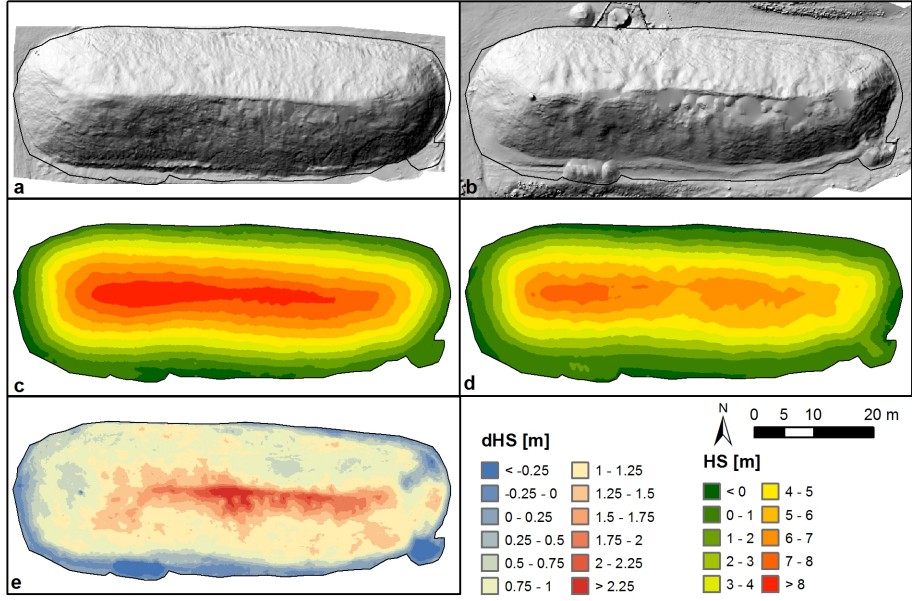

**Figure 5.** Flüela snow heap: Hillshade images (a: 29 Apr, b: 8 Oct), snow heights HS (c, d) and snow height change dHS (e).

The calculated spring snow volume of the Flüela snow pile, including covering material was $6862\,m^3$ (Martell: $7138\,m^3$) and decreased to $5307\,m^3$ (Martell: $4820\,m^3$). This corresponds to a shrinking of 23% (Martell: 32%). If we relate volume reduction to the initial volume of snow after removing the estimated amount of covering material (Flüela: $830\,m^3$, Martell: $860\,m^3$), these values increase to 26% for Davos and 37% for Martell.

Nevertheless, it needs to be considered that volume loss and dHS are not only attributed to snow ablation but also to densi-fication of the snow by settling. The density measurements described above point at a contribution of about 9%. This would mean that the effective snow ablation reduces to 15% (28%) or – in other words that 72 to 85% of the snow mass that had been covered in spring could be conserved over the summer. Note that some snow will additionally be lost during de-covering and distribution of the snow to the ski tracks.

Results obtained from measurements are compared to results of the SNOWPACK simulations in the next section. It is, however, not meaningful to relate model results calculated for a single point to volume changes for the entire snow pile. We therefore chose to relate model results to the respective coordinate (and settings of the model run) which are the locations of maximal HS. A quadratic area of $1\,m^2$ surrounding this point was used for the calculation of measured dHS. At these areas the pile shrank by 1.35 m (standard deviation = 2.5 cm) at Flüela and by 1.25 m (standard deviation = 2 cm) at Martell

corresponding to a relative reduction of $15.1 \pm 0.3\%$ (Martell: $16.6 \pm 0.2\%$).





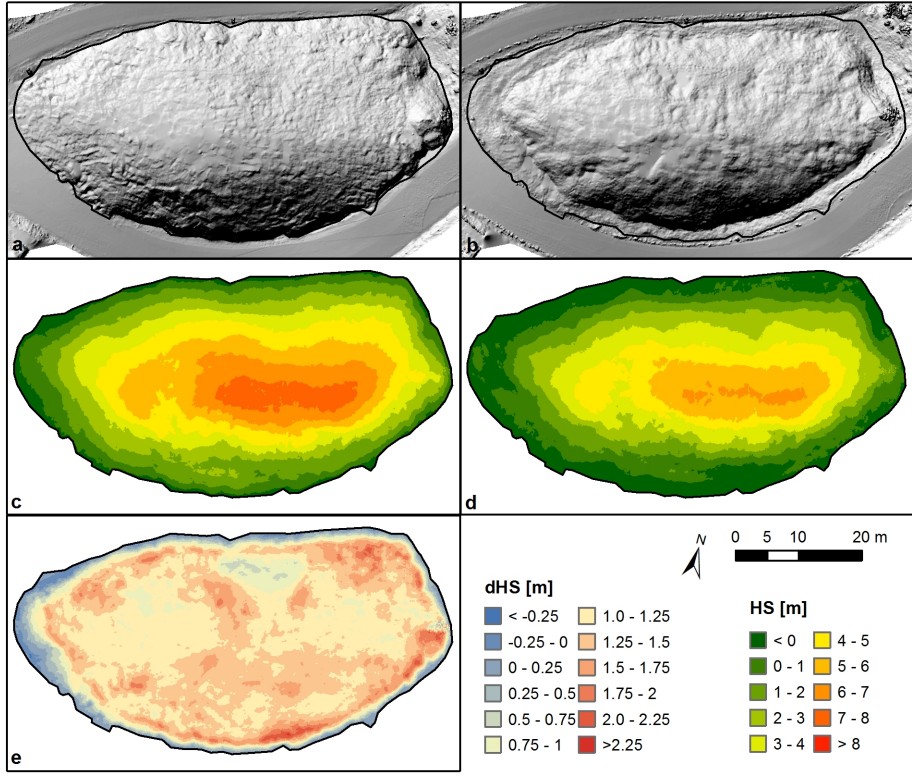

**Figure 6.** Martell snow heap: Hillshade images (a: 19 Mai, b: 28 Oct), snow depth HS (c, d) and snow height change dHS (e). Displayed are raw data that were not corrected for remaining snow in the depot (c, d) and fresh snow at the surface (d, e).

## 3.4 Model results

Figure 8 illustrates the temporal evolution of HS and density of the 40 cm sawdust covered snow heap in the Flüela valley. The pronounced yellow line parallel to the surface reflects the lower boundary of the sawdust layer. The initial density of this layer ($323\,kg/m^3$) rapidly increased in the first three weeks due to rising water content caused by rain (initial value 20%). Afterwards

5  density and water content remained at a relatively constant level of about $580\,kg/m^3$ and 45% respectively meaning that the sawdust was always wet. Higher values cannot be reached due to a threshold in the model settings of the soil layer as described in Sect. 2.4. Liquid water fraction of the snow did not change considerably in relation to the initial 3%. This is attributed to the maximum water storing capacity of a snow texture which is implemented in SNOWPACK as derived from Lütschg (2005).

The height of the heap decreased steadily from 9 m to 7.60 m. This corresponds to a relative decrease in HS of 15.6% and

10  accords extremely well with the result obtained from the measurements at the point of maximal HS. Results of the reference simulation are very similar for the Martell snow pile (not shown): Height decreases by 17.1% from 7.60 m to 6.30 m which is again an excellent match with the measurements.





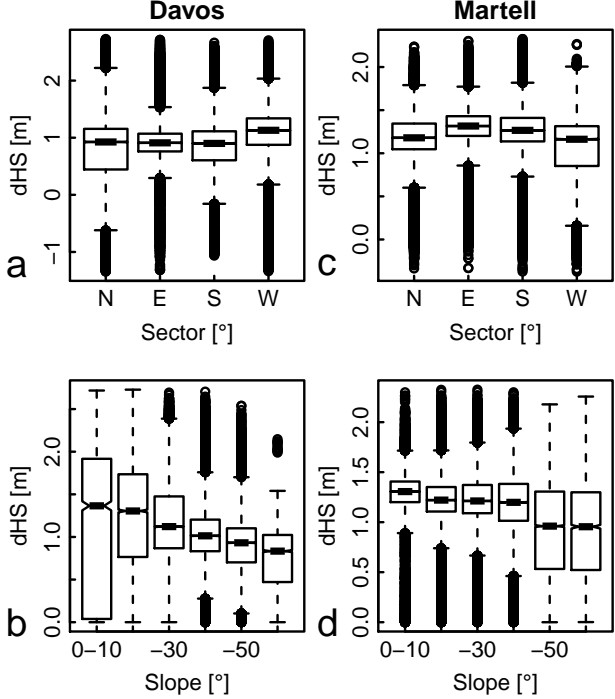

**Figure 7.** Boxplots relating dHS to exposition (a,c) and slope (b,d) of the spring survey for Flüela (a,b) and Martell (c,d)

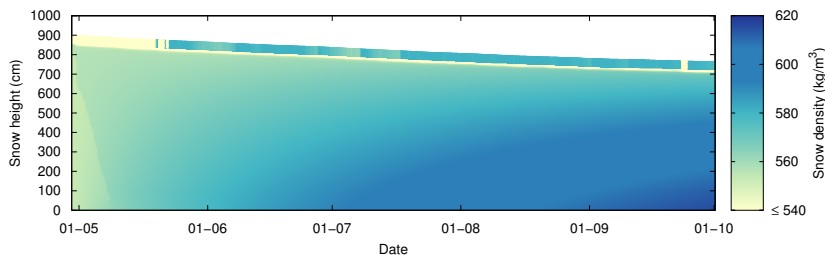

**Figure 8.** Simulated temporal evolution of density at the Flüela snow heap.

Figure 8 also shows densification. It is well illustrated how snow is compacted from top to bottom and in the course of time. While simulations were initiated with a constant density of $553\,kg/m^3$, density at 8 Oct ranged from $570\,kg/m^3$ at the top to $617\,kg/m^3$ at the bottom of the profile. A comparison with the measurements presented in Section 2.2.2 denotes that the model overestimated snow density in the upper sections of the profile and underestimated it near to the bottom. The simulated bulk density of the entire profile was $600\,kg/m^3$. Hence, snow densified by 8% during the summer season. These values agree



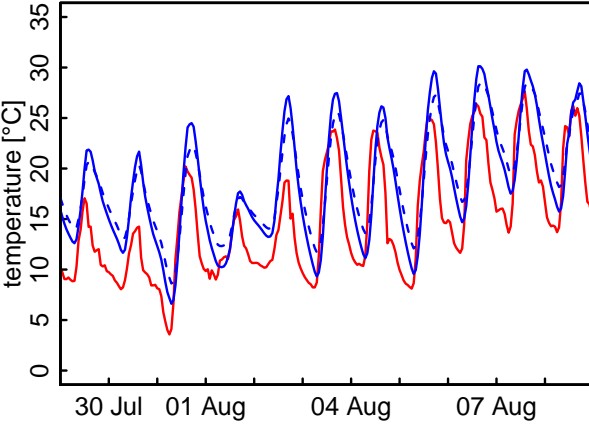

**Figure 9.** Air temperature (red) and surface temperature simulated for the Fluela snow heap with 40 cm (solid blue) and 20 cm (dashed blue) sawdust layer during four days in August.

well with the mean density calculated from the measurements ($606\,kg/m^3$, 9%). Surface temperature (TS) varied between 0 and 33°C for the reference scenario and showed a diurnal variation in the range of 5 to 20°C (Fig. 9).

### 3.5 Sensitivity study

Figure 10 summarizes volume and mass losses of snow in dependence of the different settings (Table ) analyzed in the sensitiv-
ity runs. Black dotted lines present dHS and white triangles changes in snow water equivalent (dSWE). The difference pictures densification.

Figure 10a illustrates the influence of the depth of the sawdust layer. While the entire snow heap melted by mid of September when no covering layer was set, dHS and dSWE respectively reduced to 15.5% (dSWE: 7.2%) for the 40 cm deep sawdust layer (reference) and finally to 11.1% (dSWE: 1.5%) for a 60 cm deep cover. The curves are characterized by an exponential decrease
and are clearly flattening at a thickness of about 40 cm. Reasons are that thicker layers decrease temperature gradients between snow surface and atmosphere. Doubling the height of sawdust, for example, reduces the gradient by a factor of two. As a result, energy input to the snow diminishes as more energy can be stored in the larger sawdust volume and thermal conduction to the snow decelerates. In addition, thicker covers add to dampening effects on TS as illustrated in Figure 9. Amplitude and extrema, especially minima are clearly enhanced for shallower covering layers resulting in higher TS during days and remarkably cooler
TS during nights. Heating of the sawdust during the day and cooling during the night appears delayed by few hours. This results in regular changes of the direction of the energy fluxes at the saw-dust surface. On average, however, TS (13.8°C for 40 cm sawdust) appeared clearly warmer than TA (11.3°C) resulting in a net negative heat fluxes of sensible heat and longwave radiation. This mean temperature difference between surface and air clearly reduces for smaller sawdust heights (11.8°C for 20 cm), consequently diminishing cooling effects by sensible flux and longwave emission (Table 5). Once a cover layer is thick
enough to prevent the surface temperature from dropping down significantly underneath the air temperature, the insulation of





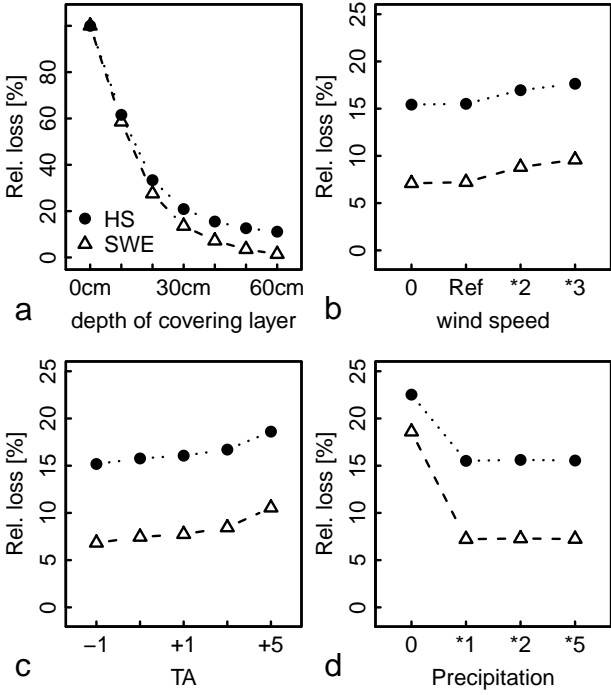

**Figure 10.** Relative loss of snow height (HS) and Snow water equivalent (SWE) calculated by SNOWPACK simulations with different thickness of the sawdust layer (a), different wind speeds (b), air temperatures (c) and precipitation (d).

a snow heap works well. Having enough capacity (mass, specific heat) to store the energy during the day while not conducting much to the snow and releasing energy efficiently during night can then be assured. Increasing layer thickness upon a certain limit shows only a minor improvements of insulation (Fig. 10a). We assume that temperature gradients in the deeper parts of the cover are then no more affected by daily changes. Moreover, the relative increase of the layer decreases with thicker covers.

Finally, Figure 10a indicates that a depth of about 30 to 40 cm is required to reach volume savings of 20 to 30% while the effect of additional sawdust is minor.

The contribution of the different terms of the energy balance at the surface of the heap is shown in Figure 11a and Table 5, where positive values designate a flux towards the heap (energy source) and negative fluxes a direction to the atmosphere (energy sink). Note that the terms as presented here are net values cumulated over the entire simulation period. The huge

effect of the covering layer on snow ablation is best illustrated when comparing the energy balance of the reference simulation (40 cm, 5th column in Fig. 11a) to a run without any cover (1st column). In total, energy available for ablation is nearly twelve times higher for the simulation without cover. In detail, shortwave radiation is by far the largest source of energy (Table 5, Fig. 11). Due to the much higher albedo of pure snow, net shortwave radiation reduces by about $1/5$ without sawdust. Contrary, longwave radiation acts as energy sink for both simulations but is nearly 13times higher when a 40 cm sawdust cover

is present. All other terms of the surface energy balance, namely net sensible and latent heat fluxes and precipitation differ in





**Table 5.** Net energy fluxes at the surface of the snow heap summed for the entire simulation period without and with a 20 and 40 cm sawdust layer.

| Height of covering layer | 0 cm | 20 cm | 40 cm |
|---|---|---|---|
| Shortwave radiation [$kJ/m^2$] | 336.5 | 410.0 | 410.0 |
| Sensible heat [$kJ/m^2$] | 74.2 | -50.0 | -73.1 |
| Latent heat [$kJ/m^2$] | 23.5 | -26.0 | -22.7 |
| Longwave radiation [$kJ/m^2$] | -21.5 | -237.6 | -275.6 |
| Precipitation [$kJ/m^2$] | 5.4 | -1.7 | -2.9 |
| Sum [$kJ/m^2$] | 418.0 | 95.1 | 35.5 |

sign: While they contribute to melting without cover, they remarkably cool the sawdust-covered snow heap and therefore limit snow ablation. The highest effect is clearly attributed to longwave emission, but sensible and latent heat fluxes also contribute remarkably. Precipitation only plays a minor role.

The cooling effect of the latent flux is mainly attributed to sublimation at the moist surface of the sawdust. As this layer remained wet for the entire summer and for all simulations shown in Fig. 10a, sawdust thickness appears less relevant for the magnitude of this flux.

Finally, Fig. 11a points out that rain, even though only marginally, adds to cooling of the heap, at least when the covering layer exceeded 20 cm. This can again be explained by temperature differences between the warmer sawdust layer and the colder rain.

In addition effects of changed atmospheric forcing, namely VW (Fig. 10b, 11b) TA (Fig. 10c, 11c) and P (Fig. 10d, 11d) have been analyzed. VW is an important parameter especially affecting turbulent fluxes (Schlögl et al., 2016). Figure 10b shows that higher VW slightly altered snow ablation. Triplicating VW, for example, resulted in an increase of dHS from 15.5 to 17.6%. Setting the wind to zero that is equivalent to a constant wind speed of 0.3 m/s, as hardcoded in SNOWPACK, reduced dHS to 15.4%. As expected, rising TA also altered snow ablation (Fig. 10c). For example, an increase in temperature of 5°C added 25 cm or 3% in dHS. On the contrary, additional rain did not affect snow ablation significantly. Even five times higher amount of rain did not change melt considerably. The reason is the wetness of the sawdust layer that never dried as already described earlier. Switching of P completely, however, increases the snow loss considerably. Smaller difference between HS and SWE for the simulation without P can be explained by less densification of the snow due to reduced moisture content in the snow pack.

In conclusion Figure 10 and 11 clearly show the effects of the covering layer on the one hand and of the atmospheric forcing on the other hand. This underlines the high correlation between sawdust thickness and energy available for snow melt. Higher VW and warmer TA also altered snow melt significantly while additional P did not play a decisive role.




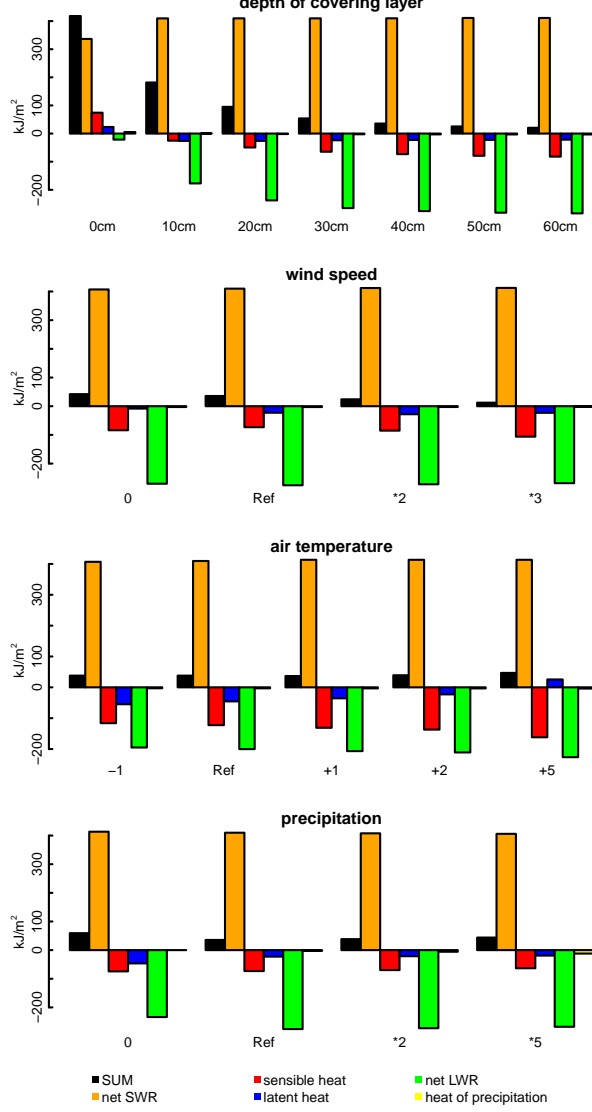

**Figure 11.** most important terms of the energy balance of the Flüela snow heap and their contribution to melt for the different model runs of the sensitivity study (a: depth of covering layer, b: wind speed, c: air temperature, d: precipitation).





## 4  Discussion

### 4.1  Data quality of the measurements and resulting error estimates

In general, we rate the accuracy of the snow volumes calculated from TLS data as very high. As described before (Sect. 2.2.1),
short measurement distances, convenient scan angles, high point densities, and overlapping of multiple scan positions provide

favorable conditions for a high accuracy of the measurements. Based on operating experiences with similar settings we esti-
mate the vertical accuracy of the TLS measurements to about 1 cm. Nevertheless, scan shadows still caused some data gaps,
especially at the crown of the heaps for the two surveys in autumn (Fig. 5b, 6b). These data gaps were determined by a rougher
surface such as local depressions. They had therefore to be closed by linear interpolation introducing some uncertainty. The
extent of the gaps is, however, limited to few square meters, meaning that the effect on the total mass balance can be rated as

marginal.

Another source of potential error is introduced by the lack of (accurate) bare ground elevation models. For Flüela no such
model could be monitored but the flat and only slightly sloped ground area allowed for a good approximation with a sloped
plane defined by the margins of the snow heap. For Martell, a bare ground elevation model was measured after most of the
snow had been distributed in autumn. The remaining snow in the depot, however, could only be estimated based on visual

impression and the rating of the local expert. As described in Sect. 2.2.1 we assumed snow volume to be in the range of 600
to 1000 $m^3$ and used 800 $m^3$ for the corrections. Applying the maximum or minimum estimates would reduce/alter relative
snow volume loss by one to two percent. Nonetheless, this correction only affects snow volumes. Snow volume changes or
dHS calculations do not require a bare ground elevation model and are therefore not concerned. Finally, thickness of sawdust
and fresh snow were obtained from a limited number of probe measurements (see Sect. 2.2.1). Nevertheless, a bias of few

centimeters would be small regarding its relation to the large volume and HS of the snow heaps.

When analyzing SWE instead of HS or volume, uncertainty in snow density measurements must also be considered. We
showed that the range of snow densities was considerable (541 to 681 $kg/m^3$) with higher densities near to the ground.
Throughout the storage period a load and time dependent densification must be assumed due to creep and wet snow metamor-
phism. As in spring only densities near to the surface could be measured, an adequate initial density profile with an expectable

increase with depth could not be captured. This limits the capability to assess the temporal evolution of densification based on
measurement but also from the simulation results as the initial snow profile had to be defined with a constant density.

### 4.2  Comparison of the sites

As described earlier, the results of the snow-volume measurements suggested large differences in terms of snow-volume loss
between the two sites. While only 22,6% of the volume disappeared at Flüela, the decrease was more than 10% larger at

Martell (Table 4). Considering the large similarity in terms of initial volume, surface area (Table 4) and also meteorological
conditions (Fig. 3, 4) this huge difference appears surprising. Possible explanations are different properties of the covering
materials (see Sect. 2.1). If we, however, consider the very similar dHS at the position of maximal HS (15% at Flüela and 16%
at Martell) and the fact that the simulations reproduced this amount very well, this explanation appears insufficient. A second





possible reason is a potential warming effect of the black paved road at Martell (Fig. 2) resulting in lateral advection of heat as described earlier (Mott et al., 2013). The relatively high dHS near to the edges of the heap (Fig. 6) supports this hypothesis. Other mico-meteorological characteristics, such as deviating local wind fields and their implication for energy balance might be present and could well play a role. Unfortunately, due to lack of measurement stations directly at the heaps, such effects

could not be detected and therefore not proofed.

### 4.3 SNOWPACK

The excellent agreement of measurements and model results indicates that SNOWPACK is well capable to reproduce dHS of sawdust covered snow heaps at the point scale. A direct transferability of these results to total mass loss of the heaps happening in the three dimensional space requires caution. As shown in Section 3.3 the total mass loss was much higher than the loss

measured and simulated at the point scale. Spatial effects such as variability of radiation and wind and lateral effects at the edges of the heap would need to be considered to simulate total mass losses. Distributed models such as Alpine3D (Lehning et al., 2006) could be used for this task.

In principal, it needs to be considered that SNOWPACK incorporates some simplifications that might influence simulation results. Such, the sawdust cover represented as homogenous layer, discriminated to eight elements in the model. Potential

internal heterogeneity such as variable temperature or water content in the layer and its implication to energy balance are not considered. Furthermore, model outcomes are strongly influenced by the accuracy of the input parameters. For example, small-scale variability in meteorological conditions might be present at the test field but not be represented in the recorded meteorological measurements. Moreover, several initial settings such as sawdust properties and the initial state of the snow pack are only based on estimations or little measurements (see Sect. 2.2). However, sensitivity runs (not shown) that had been per-

formed for the most important parameters (thermal conductivity, water content, texture of sawdust) showed no significant effect on the mass balance. Initial snow pack characteristics (density, water content) were more important but still only marginally changed final results.

### 5   Conclusions and Outlook

A detailed study on snow farming for touristic applications has been presented. Mass balances of two snow heaps covered

with a 40 cm thick layer of sawdust and chipped woods respectively have been calculated from repeated TLS surveys in 2015. More than 75% of the snow volume of a $7000\,m^3$ snow pile could be conserved in the Flüela valley near Davos, Switzerland (Table 4). At the Martell heap, only two thirds of the snow remained in autumn, even though settings and conditions were quite similar at both study sites. We assume that this reduced performance is attributed to heat advection from a surrounding paved road. Moreover, we applied the physically based snow cover model SNOWPACK to simulate snow ablation at the point scale.

A comparison of measurements and simulation results showed excellent agreement: At Flüela dHS of 15.1% (Martell: 16.6%) was measured at the point of maximal snow height. Simulations of relative losses (15.6% for Flüela and 17.1% for Martell)





were only marginally higher. In summary, the magnitude of these results is well in line with operating experiences of different snow farming sites.

Snowpack simulations were also applied to analyze the contribution of the different terms of the energy balance to snow ablation (Fig. 11). It could be shown that shortwave radiation was by far the most important source of energy. Sensible and

latent heat fluxes also contribute to melt if the snow heap was not protected by sawdust. The presence of such a covering layer, however, led to an inversion of the seasonal net fluxes of sensible and latent heat now contributing to cooling of the snow heap. The largest cooling effect was attributed to longwave emission at the surface of the sawdust. This insulting layer absorbs the solar energy during the day but strongly limits conduction to the snow. During night the absorbed energy is then emitted by longwave radiation. Such, about 2/3 of the net energy input by solar radiation is compensated by long wave cooling.

Additional 18% are balanced by net sensible and 6% by latent heat fluxes. In summary, only 9% of the net shortwave energy input remained available for snow ablation. The high amount of snow conserved over the summer is therefore attributed to these cooling and insulating effects of the covering layer. Moreover, the high influence of the thickness of the sawdust was evident from the simulations. The larger the covering layer, the smaller the temperature gradient and such the heat flux into the snow. This increased insulation effect finally results in the higher amount of snow that can be conserved. The excellent insulation

of sawdust is primary given by its high density and heat capacity rather than by its heat conductivity (factor two higher than conductivity of PP-insulation plate). This enables the sawdust to absorb large amounts of heat being re-emitted during night. Still, significant costs of sawdust and additional work load need to be considered when deciding about layer thickness. 40 cm seems a good compromise.

Finally, effects of varying meteorological conditions, namely VW, TA and P have been investigated. As expected, it could

be shown, that higher TA and VW enhance snow ablation. Nevertheless, a resulting increase of a few percent appears small in relation to the effect of covering layer thickness. This finding points out that snow farming might also be feasible under much warmer climatic conditions indicating that it might also be applicable for lower situated, sub-alpine ski resorts and communities. Additional precipitation did not play a significant role, as the sawdust remained wet during the entire summer for all scenarios with rain. Switching off rain completely, however led to a clear increase of snow ablation. As a consequence,

irrigation of snow heaps, as suggested by some practitioners, seems unnecessary as long a certain amount of rain is available.

In conclusion it could be shown that snow farming appears as appropriate method for the allocation of a basic snow offer in autumn. However, operating costs and space requirements are considerable limiting the amount of snow that can be stored. SNOWPACK proofed as appropriate simulation tool and can well be applied to study the suitability of potential snow farming sites, or to asses performance of different covering materials (see also Olefs and Lehning, 2010) or to simulate influences

of changing climatological conditions on snow farming. The only prerequisite is the availability of adequate meteorological input data. Spatial distributed models such as Alpine3D could be used to investigate spatial effects. The application of multiple different snow models could be interesting to assess model sensitivity and uncertainty. Future research might also include detailed surveys of other snow farming projects, possibly in higher temporal resolution or the investigations of mass losses during different work steps, such as shaping or de-covering of snow heaps. Investigations of different types of covering material

are currently running in Scandinavia and will contribute to knowledge about best practice of snow farming.





*Acknowledgements.* We are grateful to all people who contributed to the successful completion of this paper. Norbert Gruber, Werner Buzzi (Davos) and Martin Stocker (Martell) are acknowledged for the good collaboration during field work and for fruitful exchange of ideas. We thank Thomas Rinderer for his work, providing many basic information for the simulations. We thank Mathias Bavay and Charles Fierz for their help with SNOWPACK, Nander Wever, Hansueli Rhyner and Rebecca Mott for their support and for good discussions. Parts of this

5 study were conducted in the framework of the Swiss Competence Center for Energy Research - Supply of Electricity (SCCER-SoE) with funding from the Commission for Technology and Innovation CTI (grant 2013.0288).





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
