# Peer review of "Snow farming: Conserving snow over the summer season"

_The Cryosphere, 2017_

## Referee Comment (RC1) · J. Garvelmann (Referee) · 8 Jul 2017

Review of manuscript tc-2017-93 Snow farming: Conserving snow over the summer season by Grünewald et al.

The authors present a very interesting study about the monitoring of two snow heaps along with 1D modeling of the snow that is conserved over one summer season. Snow farming and snow management are topics that will receive more and more attention in the coming years. The focus of the presented study is in the scope of the journal. I like the study very much. However, I recommend some revisions of the manuscript prior to a publication in The Cryosphere.

The authors nicely describe the background and the motivation for snow farming. How-

ever, there is missing the clear formulation of the motivation for the study and the most important research questions that the study will address at the end of the introduction section.

The methods are described clear and understandable. On page 7 (lines 24+25) the authors mention that they used the model SNOWPACK for the simulation of both snow heaps. However, there are only shown results from the Flüela snow heap in the results section later. I recommend showing also results from the Martell site for completeness of the study.

There are sentences that should be moved to the introduction section. The sentence on page 7, lines 25-27, for example, describes the motivation for the modeling of the snow heaps. Or the sentence on page 8, lines 22-25 describes the motivation for the sensitivity analysis.

The simulation of the stored snow was carried out using the one-dimensional snow cover model SNOWPACK. However, the authors mention twice that the use of a spatial distributed snow model such as Alpine3D for example would have been more appropriate to model the snow piles. The first thought while reading the manuscript is, why such a model was consequently not used in this study? Please provide an explanation for this.

The simulation was carried out just for one point of the snow heaps, the point with maximal HS. Please indicate those points in figure 5 and 6. Why was the simulation not carried out for multiple points at the snow heap?

Another concern is related to the used parameters shown in the results section. Why are the results shown for snow height? You have height/volume and density. Why are the simulations not carried out for SWE? Another possibility would be to calculate (and simulate) total snow mass and mass loss in kg. For the TLS measurements providing snow volume and the measured snow density it would be simple to present some quantities of total snow mass loss ect. This would also be possible for the simulations

since the calculation was carried out for a quadratic area of 1 m2 as described on page 13. Please provide the results for actual snow mass or provide at least a detailed discussion why the results are only shown in snow depth.

Page 1, Line 1: Not only for touristic purposes. The examples (Davos and Martell) you describe in your study are more for a professional sport purpose. You could mention here a range of purposes for example.

Page 1, Line 7: Please include space between the words "measurements" and "agreed".

Page 1, Line 8: "surface energy balance" would be more correct here.

Page, Line 11: of 12 instead of or 12.

Page 1, Line 13: air temperature for more clarity.

Page 2, Line 4: ..the energy balance of the snow and the glacier resulting...

Page 2, Line 10: ...contribute, but are less... Page 2, Line 23: I am not sure if the word neuralgic is appropriate here...

Page 2, Line 32: In the context of meteorological conditions, the term humiditiy is more suitable than moisture.

Page 3, Line5: for hosting instead of to hosting.

Page 3 Line 6: weather conditions.

Page 4, Line ...input for the simulations (...

Page 4, Line 20: In contrast to Flüela...

Page 4 Line 21 – Page 5, Line 2: Please revise this sentence. Redundant information.

Figure 1 and 2: The authors could think about providing a map for each study site showing the surrounding terrain.

Table 1: Is a result and should therefore be moved to the results section.

Page 6, Line 8-10: Were those measurements carried out in spring or in autum?

Page 6, Line 15: In contrast to. . .

Field measurements section: Please clearly mention here that snow temperature was not measured. The information appears some where later in the text.

Page 7, Line 7: You could introduce here the abbreviation for computer tomography mentioned later in the results section (Page 10, Line 2).

Page 7, Line 4ff: How many samples were collected and analyzed at Martell?

Page 7, Line 12: Next, . . .?

Page 7, Line 12-14: Thios information could be mentioned earlier in the description of the TLS measurements.

Page 7; Line 18: First, . . .?

Table 2: Needs a better explanation in the table caption.

Page 8, Line 1: . . .apporion of 3% liquid water.

Page 8, Line 5ff: Is this assumption really realistic that the properties of sawdust and the mixture of sawdust and wood chips are similar? I would expect that the porosity is different ect.

Page 8, Line 15: snow mass? You show snow depth in the model results. Please see also my specific comments to this aspect of the study.

Page 8, Line 18: Please update the number of the table you are referring to here.

Page 9, Line 8: Please introduce the abbreviation WT earlier.

Figure 3: You are showing net longwave, right? Please add this info. It would also be very helpful to indicate the exact dates when the snow heaps were covered with the

isolating material and when it was removed. Please provide the same figure for the Martell site as well.

Page 11, Line 2: Why is the focus on Flüela?

Table 4: Please provide this table after Figure 5 and 6.

Figure 5+6: The authors could also provide a figure with the fraction (in percent) of snow loss at the two snow heaps.

Page 14, line 9: You describe earlier that the model was initiated with 8,6 m. Please clarify.

Page 14, line 11: Same here. The max HS was described to be 7,2 m earlier in the text. Please check. It would also be very helpful to mention the actual measurements again here for more clarity,

Caption figure 8: Please revise this caption. Shown are the evolution snow height AND density. It would also be very helpful to add the actual measured values to that figure. Please provide the same figure for the Martell snow heap.

Page 15, line 2: Earlier in the manuscript you mention that snow density was 555 kg/m3. Please ckeck.

Page 16, Line 4: Please add here the number of the table.

Caption figure 9: Measured air temperature and simulated. . ..

Page 17: An explanation of figure 10b, 10c, and 10d is missing.

Page 18, Line 4: I think the term evaporation is better in this context.

Page 18, Line 21: Please quantify this high correlation here.

Page 18, Lines 7-9: This is hardly visible in figure 11. . .. I recommend to recolor the sum of the individual energy balance components and change the color of heat of precip to black.

Figure 11: Is there a reason why you just show those results for Flüela and not for Martell?

Page 21, Line 4: . . .and could also play a role. . . .stations directly on the heaps, . . .

Page 21, Line 19: limited instead of little.

Page 22, Line 2: Please provide more informationhere.

Page 22, Lines 9+13: the words "such" are a bit strange here. Please revise those sentences.

Page 22, Line 27: Please provide more information about operation costs. I think this is very important information here for interested readers.

---

## Referee Comment (RC2) · B. Nordell (Referee) · 24 Jul 2017

This is paper is scientifically good but a bit dry to read. It could be improved by moving details of the measurement equipment to an appendix. I suggest that the paper is accepted for publishing in TC after some minor editorial changes in the text.
* * *

---

## Editor Comment (EC1) · G. Chambon (Editor) · 24 Aug 2017

Dear authors,
For your information, in order to ensure a balanced and optimal reviewing process, we decided to extend the interactive discussion of your paper and give more time to additional referees to submit their comments. We apologize for the additional delay induced by doing so.
Best regards,
Guillaume Chambon / TC Topical Editor.
* * *

---

## Author Comment (AC1) · 13 Sep 2017

First we want to thank J. Garvelmann for his constructive review and his good suggestions. We are answering his comments in the following, for clarity we repeat the original comment (C) and answer (A) afterwards:

Major comments:

C: The authors nicely describe the background and the motivation for snow farming. However, there is missing the clear formulation of the motivation for the study and the most important research questions that the study will address at the end of the introduction section.

A: Thank you for this hint we add the following sentence at the end of the Introduction

section: "A rising number of expertise on snow farming has been requested at SLF in recent years. This motivated us to (i) provide a review on current snow farming praxis (ii) to perform a detailed field study on snow farming and (iii) to describe and evaluate the model used for snow farming expertise by the SLF. "

C: The methods are described clear and understandable. On page 7 (lines 24+25) the authors mention that they used the model SNOWPACK for the simulation of both snow heaps. However, there are only shown results from the Flüela snow heap in the results section later. I recommend showing also results from the Martell site for completeness of the study.

A: This is principally right, we focus on the Flüela data and only provide the most important information for Martell. The reason is that the principal findings from both sites do not differ much: Adding more Martell details would not provide new findings and the added value would be very small. For readability we therefore decided to focus on the Flüela results and to only show the most important values for Martell (in brackets). This is also described in the text.

C: There are sentences that should be moved to the introduction section. The sentence on page 7, lines 25-27, for example, describes the motivation for the modeling of the snow heaps. Or the sentence on page 8, lines 22-25 describes the motivation for the sensitivity analysis.

A: It is right that these sentences would also fit to Introduction content-wise. However, Introduction already contains this information. We deliberately repeat this information in section 2.4 for readability and clarity.

C: The simulation of the stored snow was carried out using the one-dimensional snow cover model SNOWPACK. However, the authors mention twice that the use of a spatial distributed snow model such as Alpine3D for example would have been more appropriate to model the snow piles. The first thought while reading the manuscript is, why such a model was consequently not used in this study? Please provide an explanation

for this.

A: We see the point of the reviewer. It is right that using a distributed model such as Alpine3D could account for spatial heterogeneity (most important: insolation depending slope and aspect) but spatial distributed input information (e.g. information on the local wind field, spatial variability of the cover material) required for such more sophisticated analysis was not available. This uncertainty of the input data would probably be larger than the resulting spatial variability of the results. Moreover, as insinuated in the new sentences in Introduction, SNOWPACK is the model that has been used for snow farming engineering projects by the SLF so far. Such projects aim to provide rough estimations on expected mass losses for specific sites and covering methods. Such requests can well be answered with a 1D model (SNOWPACK). Setting up and running SNOWPACK and analyzing the results is easier and more straight forward and therefore more cost effective. Our paper shows well, that SNOWAPACK is well capable for this purpose. For detailed analysis of processes and their spatial variability, we definitely aim to apply Alpine3D in future (projects to come). This will be interesting from a scientific perspective.

C: The simulation was carried out just for one point of the snow heaps, the point with maximal HS. Please indicate those points in figure 5 and 6. Why was the simulation not carried out for multiple points at the snow heap?

A: The points are now indicated in Fig 5 and 6. As explained before, the added value of multiple point simulations would be rather small.

C: Another concern is related to the used parameters shown in the results section. Why are the results shown for snow height? You have height/volume and density. Why are the simulations not carried out for SWE? Another possibility would be to calculate (and simulate) total snow mass and mass loss in kg. For the TLS measurements providing snow volume and the measured snow density it would be simple to present some quantities of total snow mass loss ect. This would also be possible for the simulations

since the calculation was carried out for a quadratic area of 1 m2 as described on page 13. Please provide the results for actual snow mass or provide at least a detailed discussion why the results are only shown in snow depth.

A: Generally it is true, that snow mass or SWE is the quantity that would be most interesting in snow farming. However, snow height is the quantity that is measured by laser scanning. Snow height and volume can be measured very accurate. Contrary, only few density measurements required to calculate SWE or mass from HS were available, adding some uncertainty for the related quantities. Moreover, snow depth is a more concrete quantity for practitioners and laymen (who are also addressed by this paper). We therefore decided to stick with snow height/volume in Sec 3.3. Furthermore, the influence of densification is already discussed in detail, most important findings (relative losses) are also provided for SWE and simulation results are also shown (e.g. Fig 10) in SWE.

Minor comments: A: we adapt most suggestions of the reviewer in the text and only answer to non-technical comments:

C: Figure 1 and 2: The authors could think about providing a map for each study site showing the surrounding terrain.

A: We have considered showing such a map. However, considering the already large number of Figures (11) and also that the character of the surrounding area can already be seen in Figs. 1 and 2 we decided against showing additional figures.

C: Table 2: Needs a better explanation in the table caption

A: The table and the caption have been changed such that the initialization should be clearer now.

C: Page 8, Line 5ff: Is this assumption really realistic that the properties of sawdust and the mixture of sawdust and wood chips are similar? I would expect that the porosity is different ect.

A: This is a reasonable doubt. From our investigations we think that the difference between the materials is much smaller than the uncertainty in the estimations of these properties and the spatial variability of the cover material (which is especially large for the mixture of chipped wood and saw dust). To test the effect of porosity (and therefore water storing capacity) we performed some model runs with varying grain sizes of the covering layer. Increasing the grain radius (from 0.1 mm to 1 mm) by a factor of 10 did not reveal any difference in the final mass loss. Only much larger grain sizes (3 mm) increased mass loss slightly. Wood chips of that size (or even bigger) exist in the Martell covering Material, but the finer particles clearly dominated. Moreover, from laboratory measurements of small samples from both heaps we found nearly identical dry densities. We therefore believe that the assumption of same properties of the covering material is appropriate. We include a corresponding sentence in the text.

C: Figure 3: You are showing net longwave, right? Please add this info. It would also be very helpful to indicate the exact dates when the snow heaps were covered with the isolating material and when it was removed. Please provide the same figure for the Martell site as well

A: Yes it is net longwave. We will clarify this in the new draft. The figure for Martell is already in the paper (Fig 4). The heaps were covered from mid of April till 19 October in Flüela and from 19 May till 28 October in Martell. Snow was then immediately distributed to the tracks. We add this information in the Study site section.

C: Figure 5+6: The authors could also provide a figure with the fraction (in percent) of snow loss at the two snow heaps.

A: This is a good suggestion, but the added value is only small and we are therefore not showing an additional figure.

C: Page 14, line 9: You describe earlier that the model was initiated with 8,6 m. Please clarify

A: 8.6 m is the height of snow without saw dust and 9 m is the height of the entire heap with saw dust cover. We think it is already clearly stated.

C: Page 15, line 2: Earlier in the manuscript you mention that snow density was 555 kg/m3. Please check.

A: 555 is the mean of the density measurements. 553 is the density used in the model. This density is calculated from the volume fractions of water, ice and void. The difference to 555 is attributed to rounding of these fractions to two digits.

C: Page 17: An explanation of figure 10b, 10c, and 10d is missing.

A: The explanation is later in the text (Page 18). A reference to Fig 10d will be added.

C: Page 18, Line 21: Please quantify this high correlation here.

A: We are not talking about a statistical correlation in that context. To clarify we change the sentence to: "This underlines the high impact of sawdust thickness on energy available for snow melt."

C: Page 18, Lines 7-9: This is hardly visible in figure 11. I recommend to recolor the sum of the individual energy balance components and change the color of heat of precip to black.

A: Figure has been recolored.

C: Page 22, Line 2: Please provide more information here.

A: We added the range of losses (12-50%) (based on an survey of several snow farming sites)

C: Page 22, Line 27: Please provide more information about operation costs. I think this is very important information here for interested readers

A: Right, so we added the following information which is based on the personal communication of the responsible persons (Norbert Gruber & Werner Putzi) of community of

Davos: "Operational costs have to be evaluated for each snow farming project specifically considering the applied technical and logistical solutions. For example in Davos 15 CHF per m3 snow were estimated for the first snow farming project in 2008. Till 2016 these costs could be strongly reduced to about 9 CHF per m3 thanks to larger snow volumes stored and improved infrastructure and work flow. Investments for structural measures at the storage location are not considered in this calculation. Two thirds of the expenses were caused by the distribution of the snow along the cross-country track followed by the removal of the saw dust (14%) and material costs for saw dust (10%) , assuming a five year operational live-time (Norbert Gruber and Werner Putzi personal communication). Generally, it can be stated that snow production costs are minor compared to covering and especially distribution costs. "

---

## Author Comment (AC2) · 13 Sep 2017

C: This is paper is scientifically good but a bit dry to read. It could be improved by moving details of the measurement equipment to an appendix. I suggest that the paper is accepted for publishing in TC after some minor editorial changes in the tex.

A: Thank you for this brief feedback. We believe that the methods section is mandatory for the paper and not very exhaustive anyway. We therefore think that is should remain in the main text.

---

## Referee Comment (RC3) · S.R. Fassnacht (Referee) · 6 Oct 2017

review of TC-2017-093

**General Comments**
This is a somewhat novel idea and I applaud the authors for using a modeling approach with field data to assess the utility of snow farming. There are no major problems with this paper and with some clarification, it will make a good contribution.

The differences in ablation between the two sites is attributed to the "potential warming effect of the black paved road at Martell resulting in lateral advection of heat (page 21 line 1)." Can this be quantified at all? This is an important point.

The writing is good, with a few instances of paragraphs that seem to short. There are a few words uses that are somewhat subjective, such as "huge" on page 20 line 31. These can be distracting. The figures are good, but could be slightly improved, such as adding section letters (e.g., Figure 11, use a. depth of covering layer) and increase the font size on axes.

**Specific Comments**
- page 1, line 11: "a factor of 12" instead off "or"
- page 1, line 15: this sentence is confusing " switching of precipitation of completely would strongly increase melt"
- page 1, lines 17-21: another citation could be the pozo de nieve (snow wells) that were extensively used in Spain well into the 19$^{th}$ century
- p2, l 2-5: While it is an old citation, it is an interesting approach to reduce mass loss of small glaciers/snowfield due to sublimation - Slaughter (1970 US Army CRREL Special Report 130)
- p3, l 27: it may be intuitive to you, but add direction to the location, i.e., lat: 46.808°N, long: 9.868°E (I assume).
- p3, l 29-30: could you simulate how different natural snow in piles would be? "A large snow pile is formed by machine made snow produced during the winter months."
- p4, l 3-5 and 12-13: did you compare the met station on top of the building to the met station on top of the snow heap?
- p4, l 4-7: you use new symbols that do not seem common – air temperature (TA), relative humidity (RH), wind speed (VW), direction (DW), incoming shortwave radiation (ISWR), incoming longwave radiation (ILWR). Are this necessary, or can you use more common symbols?
p5, l 14: you "calculate snow volumes." What about mass?
p5, l 17: state the wavelength of the TLS "near-infrared spectral range"
p7, l 10: is the word "extremely" necessary?
p7, l 19: do you mean "crown" instead of "crone?"
p7, l 19: be specific about the type of "linear interpolation"
p7, l 31: the "grain size of 1mm" seems quite small
Table 2: is a spectral albedo of sawdust of 0.5% correct? This seems low, or explain what this is.
p8, l 13-14: data were "resampled to 30 min time steps" but the "modeling time step was set to 15 min." Please rectify or discuss this discrepancy
p8, l 18 and 21: should this read "Table 3-5", instead of Table 3.5?
p9, l 15: change "Lower temperatures and irradiation **is** mainly"

Figure 3c: maybe use two axes the same as Figure 3a and b, with net SWR in red on the left and net LWR in blue on the right.

Figures 3 and 4: use Oct rather than Okt. Think about putting these two sets of figure beside one another

p11, l 3-4: delete the first two sentences: "This section presents results obtained from the TLS surveys. The focus of the analysis is on the Flüela data set. Values for Martell are provided in brackets."

p11, l 5 and beyond: use a period as the decimal place "8.99m" rather than a comma "8,99m"

Figure 5: consider changing the color ramp so that white is no change, blue is an addition and red is a less. At present it is confusing as blue can be a small gain or loss.

Figure 5: there is a scale bar. Think about adding dimensions (in x and y) to one of the figures instead so we know how big it is.

p13, l 12: perhaps show this "respective coordinate" on Figure 5a and 6a

Figure 7: do you mean "aspect" for "exposition?" The x-axis for Figures 7b and 7d are unclear

p16, l 4: what is meant by "in dependence of the different settings?" Also, add a number to "(Table )"

p16, l 4-6: I would delete these sentences. They are not necessary. What is meant by "The difference pictures densification?"

Figure 10d. I am surprised that precipitation does not appear to change the results at all

Table 5: how is the albedo of the snow heap modeled over time, as this influence net SWR.

Figure 11: would this be clearer is there were log scales (both positive and negative), as some of the bars are difficult to see.

p20, l 9-10: can you quantify "total mass balance can be rated as marginal?"

p20, l 31: is the word "huge' necessary?

p20, l 31-31: "Possible explanations are different properties of the covering materials." I assume that you did not model Martell? Can you do some simple calculations to describe the differences between saw dust and wood chips. While you subsequently say that it is likely not important, we don't know this.

p21, l 1: can you try to quantify the "potential warming effect of the black paved road at Martell resulting in lateral advection of heat?"

p21, l 14: delete "such"

p22, l 28: I think you mean "proved" rather than "proofed." I think that this word is too strong as you only modeled one point at the top of the pile.

---

## Author Comment (AC3) · 8 Nov 2017

We are grateful to S.R. Fassnacht for his very positive and constructive review. We are answering his comments in the following, for clarity we repeat the original comment (C) and answer (A) afterwards:

General Comments

C: This is a somewhat novel idea and I applaud the authors for using a modeling approach with field data to assess the utility of snow farming. There are no major problems with this paper and with some clarification, it will make a good contribution. A: Thank you for this positive feedback.

C: The differences in ablation between the two sites is attributed to the "potential warm-

ing effect of the black paved road at Martell resulting in lateral advection of heat (page 21 line 1)." Can this be quantified at all? This is an important point. A: Together with the good suggestion to do full three-dimensional (Alpine3D) simulations also for the snow heap, it will be interesting to see (probably in a future study), in how far the road may contribute to stronger melt. However, it will not be adequate to present a calculation of the effect in the current paper, which presents a simplified analysis based on one-point simulations at the top of the pile. A "back of the envelope calculation", which could be done based on an assumed road temperature could be done but is considered not to be very useful as we don't have time-resolved estimates of surface temperatures for both the side of the pile and the road. Thus, at least rudimentary radiative transfer modelling in combination with local energy balance modelling would be required to quantitatively estimate the effect. For this, we do not have sufficient input data and it is beyond the scope of the analysis as presented in this paper.

C: The writing is good, with a few instances of paragraphs that seem to short. There are a few words uses that are somewhat subjective, such as "huge" on page 20 line 31. These can be distracting. The figures are good, but could be slightly improved, such as adding section letters (e.g., Figure 11, use a. depth of covering layer) and increase the font size on axes. A: We are carefully revising these points and implementing changes where required (see detailed comments)

Specific Comments

- page 1, line 11: "a factor of 12" instead off "or" A: changed

- page 1, line 15: this sentence is confusing "switching of precipitation of completely would strongly increase melt" A: Sentence was changed to "No significant effect of additional precipitation could be found as the sawdust remained wet during the entire summer, already with the measured quantity of rain. Setting precipitation amounts to zero, however, strongly increased melt."

- page 1, lines 17-21: another citation could be the pozo de nieve (snow wells) that

were extensively used in Spain well into the 19th century A: Thank you for this hint, we have added a citation (Morley 1942)

- p2, l 2-5: While it is an old citation, it is an interesting approach to reduce mass loss of small glaciers/snowfield due to sublimation - Slaughter (1970 US Army CRREL Special Report 130) A: It is indeed an interesting paper. As it is, however, not directly linked to snowfarming we believe that it is not meaningful to discuss in the context of our study.

- p3, l 27: it may be intuitive to you, but add direction to the location, i.e., lat: 46.808°N, long: 9.868°E (I assume). A: We have added directions for both study sites.

- p3, l 29-30: could you simulate how different natural snow in piles would be? "A large snow pile is formed by machine made snow produced during the winter months." A: This would be possible by changing the initial properties of the snow heap in the input file. However, differences between well settled, aged natural and technical snow are marginal and simulation results with old natural snow would therefore not reveal significant differences.

- p4, l 3-5 and 12-13: did you compare the met station on top of the building to the met station on top of the snow heap? A: Yes we did for TA and VW and developed a correction for VW. This is described in Sect. 3.1

- p4, l 4-7: you use new symbols that do not seem common – air temperature (TA), relative humidity (RH), wind speed (VW), direction (DW), incoming shortwave radiation (ISWR), incoming longwave radiation (ILWR). Are this necessary, or can you use more common symbols? A: We changed symbols for wind speed (WS) and wind direction (WD).

p5, l 14: you "calculate snow volumes." What about mass? Snow mass or SWE cannot be directly calculated from TLS measurements. It also requires snow density. Only few density measurements were available, it would therefor add uncertainty to results to use SWE instead of HS (which was measured highly accurate). We therefore focus

on volume for the measurement but also address SWE when analysing model results (also see reply to review by J. Garvelmann).

p5, l 17: state the wavelength of the TLS "near-infrared spectral range" A: the wavelength is 1064 we added it in brackets.

p7, l 10: is the word "extremely" necessary? A: removed

p7, l 19: do you mean "crown" instead of "crone?" A: yes that's right.

p7, l 19: be specific about the type of "linear interpolation" A: We changed to ". . .by triangulation with the nearest points".

p7, l 31: the "grain size of 1mm" seems quite small A: This should be "grain radius" instead of grain size. 2mm is a typical grain size for well settled technical snow.

Table 2: is a spectral albedo of sawdust of 0.5% correct? This seems low, or explain what this is. A: yes, this is a mistake. It should be 50%Âĺ

p8, l 13-14: data were "resampled to 30 min time steps" but the "modeling time step was set to 15 min." Please rectify or discuss this discrepancy A: We changed to "All input data were filtered, quality checked and resampled to the modeling time step of 15 min using the meteorological input-output library MeteoIO"

p8, l 18 and 21: should this read "Table 3-5", instead of Table 3.5? A: it should be Table 3.

p9, l 15: change "Lower temperatures and irradiation is mainly A: changed

Figure 3c: maybe use two axes the same as Figure 3a and b, with net SWR in red on the left and net LWR in blue on the right. A: I think that the figure is quite clear as it is. I see no reason for adding a second axes with the same units and dimensions.

Figures 3 and 4: use Oct rather than Okt. Think about putting these two sets of figure beside one Another A: labelling has been changed.

p11, l 3-4: delete the first two sentences: "This section presents results obtained from the TLS surveys. The focus of the analysis is on the Flüela data set. Values for Martell are provided in brackets." A: Sentences have been removed.

p11, l 5 and beyond: use a period as the decimal place "8.99m" rather than a comma "8,99m" A: changed

Figure 5: consider changing the color ramp so that white is no change, blue is an addition and red is a less. At present it is confusing as blue can be a small gain or loss. A: We changed the colour map according to the suggestion.

Figure 5: there is a scale bar. Think about adding dimensions (in x and y) to one of the figures instead so we know how big it is. A: We removed the scalebar and indicate scale in the x-axes now

p13, l 12: perhaps show this "respective coordinate" on Figure 5a and 6a A: We are showing the coordinate in Fig 5c, 5d, 6c and 6d now

Figure 7: do you mean "aspect" for "exposition?" The x-axis for Figures 7b and 7d are unclear A: yes it should be aspect. We changed it and reworked the axis.

p16, l 4: what is meant by "in dependence of the different settings?"?" Also, add a number to "(Table )" A: Deleted. See below.

p16, l 4-6: I would delete these sentences. They are not necessary. What is meant by "The difference pictures densification?" A: Deleted. It was meant to explain the difference in relative losses in SWE and HS; this is attributed to an additional effect of densification on HS (losses in HS are larger than in SWE).

Figure 10d. I am surprised that precipitation does not appear to change the results at al. A: The reason is that the cover remains wet even for the case of no additional rain. Additional wetness seems to have not much additional cooling effect. This is described in the text.

Table 5: how is the albedo of the snow heap modelled over time, as this influence net SWR.

A: Albedo of saw dust albedo is modelled with a constant value of 0.5; We cannot quantify temporal change of saw dust albedo but is probably limited over a single summer. Albedo of snow (only relevant for simulation without sawdust cover) chages in time dependent of snow properties as described in Schmucki et al. 2014

Figure 11: would this be clearer is there were log scales (both positive and negative), as some of the bars are difficult to see. A: We have tested this suggestion. As expected, log scale improves readability of the small fluxes. However, the large difference in contribution of the single fluxes to melt is then less pronounced. We think that emphasising this relative difference is more important than readability of the smaller, relatively unimportant fluxes and therefore remain with a linear scale.

p20, l 9-10: can you quantify "total mass balance can be rated as marginal?" A: yes of course; let's assume a total size of the gaps of 10m2 (which is large) and a mean error introduced by interpolation of 0.5 m (which is also large). This results in a volume uncertainty of 5m3; relating this to snow volume of the entire heap (5000-7000 m3) is below 0.1%

p20, l 31: is the word "huge' necessary? A: removed

p20, l 31-31: "Possible explanations are different properties of the covering materials." I assume that you did not model Martell? Can you do some simple calculations to describe the differences between saw dust and wood chips. While you subsequently say that it is likely not important, we don't know this.

A: We also modelled Martell, but with the same properties of the covering layer. We believe that the general properties of saw dust and the mixture of saw dust and wood chips in Martell are similar. To test this we did some modelling (see our reply to J. Garvelmanns review below). However, the large heterogeneity of the surface, especially in Martell, with very dark areas consisting of older cover material and with areas of fresh, brighter material might well have effects, e.g. on albedo.

Reply to J. Garvelmann: From our investigations we think that the difference between the materials is much smaller than the uncertainty in the estimations of these properties and the very small-scale spatial variability of the cover material (which is especially large for the mixture of chipped wood and saw dust). To test the effect of porosity (and therefore water storing capacity) we performed some model runs with varying grain sizes of the covering layer. Increasing the grain radius (from 0.1 mm to 1 mm) by a factor of 10 did not reveal any significant differences in the final mass loss. Only much larger grain sizes (3 mm) increased mass loss slightly. Wood chips of that size (or even bigger) exist in the Martell covering Material, but the finer particles clearly dominated. Moreover, from laboratory measurements of small samples from both heaps we found nearly identical dry densities. We therefore believe that the assumption of same properties of the covering material is appropriate. We include a corresponding sentence in the text.

p21, l 1: can you try to quantify the "potential warming effect of the black paved road at Martell resulting in lateral advection of heat?" A: See detailed explanation above

p21, l 14: delete "such" A: done

p22, l 28: I think you mean "proved" rather than "proofed." I think that this word is too strong as you only modeled one point at the top of the pile. A: we changed to "proved"